# Can an Aerobic Exercise Programme Improve the Response of the Growth Hormone in Fibromyalgia Patients? A Randomised Controlled Trial

**DOI:** 10.3390/ijerph20032261

**Published:** 2023-01-27

**Authors:** Nerea C. Estrada-Marcén, Jaime Casterad-Seral, Jesus Montero-Marin, Enrique Serrano-Ostáriz

**Affiliations:** 1Faculty of Health and Sport Sciences, University of Zaragoza, 22002 Huesca, Spain; 2Department of Psychiatry, Warneford Hospital, University of Oxford, Oxford OX3 7JX, UK; 3Teaching, Research & Innovation Unit, Parc Sanitari Sant Joan de Déu, 08830 Sant Boi de Llobregat, Spain; 4Consortium for Biomedical Research in Epidemiology & Public Health (CIBER Epidemiology and Public Health-CIBERESP), 28029 Madrid, Spain; 5Faculty of Medicine, University of Zaragoza, 50009 Zaragoza, Spain

**Keywords:** fibromyalgia, exercise, GH, quality of life, aerobic capacity, body composition

## Abstract

Downgrade alterations in the growth hormone (GH) might be involved in the development of some of the fibromyalgia syndrome (FMS) symptoms. Our aim was to assess the effects of an aerobic exercise programme on the GH levels in patients with FMS. A randomised controlled trial was developed. Sixty-four Spanish women with FMS were randomly assigned to the experimental arm (n = 33) and treated with a 16-week group physical exercise programme based on low impact aerobic dance (three weekly sessions, one-hour each), or to the treatment-as-usual (TAU) control arm (n = 31). The primary outcome was the GH response to acute exercise. Secondary outcomes were GH basal, sensitivity to pain, body composition, aerobic capacity, and quality of life. The ANCOVA results showed a moderate effect of treatment improving the GH response to acute exercise. Other effects were substantial for aerobic capacity, quality of life, and body composition. Pre-intervention GH response to acute exercise was related to improvements in aerobic capacity and quality of life. An aerobic exercise programme may improve the response of the GH, aerobic capacity, body composition, and quality of life in women with FMS. The normalization of neuro-hormonal patterns involving the GH might be key for improving some FMS symptoms.

## 1. Introduction

Fibromyalgia is described as a chronic-pain syndrome that can be characterized by the presence of symptoms like widespread musculoskeletal pain, stiffness, sleep disturbances, fatigue, and cognitive difficulties [1,2,3]. Fibromyalgia syndrome (FMS) is considered as a health problem due to its relatively high prevalence, affecting approximately 2% of the general population, and is often concomitant with other medical disorders such as gastrointestinal diseases, other pain-related conditions as well as mental health disorders [2,3,4], causing a considerable effect on patients’ quality of life. The diagnosis and treatment of FMS involves many specialists due to the limited knowledge of its specific aetiology, which so far remains unknown. In general, it has been suggested that social, psychological, and somatic factors may interact and have a role in the predisposition and perpetuation of FMS symptoms [2]. Different types of interventions have demonstrated certain degrees of efficacy for reducing FMS symptomatology, although most of them with small effect sizes and limited effectiveness [5]. The lack of a clear standard treatment together with the many different types of interventions that have been proposed, which may or may not be effective, make FMS as an important public health target [6].

Different etiopathogenic theories have attempted to explain the origin of FMS, although recently there is increasing evidence of alterations in the central nervous system [7,8,9,10]. The similarity between the symptomatology suffered by patients with FMS and by individuals with alterations of the Hypothalamic-Pituitary-Adrenal axis (HPA) was the factor that triggered research around this axis [11]. Several studies have tried to explain FMS based on an HPA axis dysfunction model, where the failure of the homeostatic systems, due to situations of permanent stress, may lead to chronic pain with no apparent organic causes [12]. Since the first studies performed on the HPA axis and FMS, evidence of alterations in its functioning have been found [13,14,15]. Several more recent studies have analysed possible endocrinological alterations in FMS [16,17], and currently, researchers’ attention has been focused on the role of the growth hormone (GH). It appears that the production of the GH is more reduced in patients with FMS [18,19,20], and the usual increase that occurs as a response to stress and exercise is especially affected [21].

GH is secreted by the anterior pituitary gland, producing central and peripheral effects that modulate the activity of neuronal populations involved in the control of metabolism and stress response, helping the organism restore homeostasis [22]. It has been suggested that alterations in GH values are involved in the development of some of the FMS symptoms [20]. In addition, it has been observed that individuals who present alterations in the GH and patients with FMS have a similar set of symptoms [11]. Therapy with GH in FMS has proved to improve muscle tiredness, exercise capacity, or intolerance to cold, and if extended over time, the number of tender points also seems to decrease [23]. There has also been evidence of the effectiveness of this hormone when administered to treat pain in FMS [24]. A recent study found that training through isokinetic exercises improved pain and GH levels in soccer players with chronic low back pain [25]. All these data suggest the potentially important role that this hormone might play in the development of symptomatology in FMS, and that it could be enhanced through physical exercise.

In this context, the main interest of the present study was to assess the effects of a low impact aerobic-based, high intensity, physical exercise programme, lasting for 16 weeks, on the GH levels in patients with FMS. Attention was centred on the response of this hormone to exercise, although we were also secondarily interested in the repercussion of this programme on the GH basal levels, sensitivity to pain, anthropometric and body composition measures, aerobic capacity, and quality of life. We also explored the potential relationships between the levels of GH and the rest of the variables, as well as the potential predictive value of GH response to exercise at pre-intervention, and improvements throughout the programme in the rest of variables, with the premise that the GH behaviour could be a key factor in the development of FMS symptomatology. We mainly hypothesised that the application of an exercise programme such as that described could improve the response of the GH in patients with FMS, as well as their sensitivity to pain, anthropometric and body composition measures, aerobic capacity, and quality of life.

## 2. Materials and Methods

A randomised controlled trial (RCT) was carried out. Participants were assessed before and after the physical exercise training period, and therefore, measurements included pre-test, and post-test evaluations. The investigator who was involved in all the assessments was blinded to the treatment group of the FMS patients.

### 2.1. Participants

The sample was selected from patients diagnosed with FMS who were registered in the “Aragonese Association of Chronic Asthenia and Fibromyalgia”. The inclusion criteria were to have been diagnosed with FMS by a rheumatologist according to the “American College of Rheumatology” (ACR) criteria [26], to be over the age of 18 years, to show Spanish language fluency, and to sign the corresponding informed consent. Individuals participating in physical exercise programmes, or who had carried out physical activity regularly over the last six months were excluded. Submission to some type of treatment, with the exception of pharmacological treatment, was also considered a specific exclusion criterion. Finally, patients were examined to detect any sign or symptom of illness that might prevent or advises against doing physical activity.

We based our sample size calculation on an expected between group difference that would correspond to a large effect (Cohen’s d = 0.80) on the response of GH to exercise. Based on a significance level of 5% and a statistical power (1-β) = 0.80 in a two-tailed test, we needed 26 FMS patients in each group. In line with previous research in which around 85% of participating women with fibromyalgia tolerated a moderate to high–intensity aerobic exercise programme [27], we expected a maximum attrition rate in our low impact aerobic-based, high intensity, physical exercise programme of around 20%, and thus numbers were increased to obtain a total sample size of 64 participants. Therefore, 64 women with FMS were recruited and randomly allocated (using parallel assignment and a computed-generated randomisation list that was concealed by an independent researcher) into two intervention arms: (a) the treatment group, where participants followed an aerobic dance training programme that was added to the treatment as usual (i.e., exercise arm); and (b) the active control group, where participants received the treatment as usual only (i.e., TAU arm). The aerobic exercise arm was initially comprised of 33 women with an average age of 46.9 years (SD = 9.7), while the TAU arm was initially comprised of 31 women with an average age of 48.7 years (SD = 7.8).

The study protocol complied with the legal requirements in force at the time the study was carried out in Spain (RD 561/1993; BOE 1993). The RCT was conducted according to the “Initiative on Methods, Measurement, and Pain Assessment in Clinical Trials” (IMMPACT) recommendations. All procedures performed in this study were in accordance with the 1964 Helsinki Declaration and its subsequent amendments, as well as the ethical standards for exercise and sport science research [28].

### 2.2. Interventions

The exercise intervention group followed a supervised group aerobic exercise programme based on low impact aerobic dance. It lasted for 16 weeks (48 sessions total), with three weekly sessions lasting for an hour. Aerobic resistance work was carried out during each session (40 min), combined with strength-resistance work with self-select loads (six to seven exercises aimed at the main groups of muscles, using 20 repetitions), with return to calm lasting for 10 min, based on carrying out relaxation techniques and breathing exercises. The mean intensity of the aerobic phase was situated at around 80% of the maximum heart rate and was monitored with a Polar Accurex Heart Rate Monitor. The Rated Perceived Exertion was 12 points on a scale from 6 to 20 throughout the programme sessions. A progression was followed in the physical exercise programme, and measures for treating diversity were established. The aerobic exercise intervention was based upon the 2000 American College of Sports Medicine (ACSM) recommendations for maintaining and developing cardiorespiratory and muscular fitness in healthy adults [29]. However, the aerobic exercise intervention involved several considerations that could contribute to a better understanding and practice by FMS patients, owing to characteristics of their disorder, e.g., the cognitive impairments that entail difficulties in abilities such as attention and memory, as well as generalised pain and stiffness that make it hard for them to maintain the same posture for long periods of time. The programme was taught by a Physical Education Graduate (N.C.E.-M.) who had received specific training in aerobic exercise interventions and had extensive experience in conducting training groups, and who counted with the counselling and support of a multidisciplinary group of FMS experts. For more details of the training programme, see the Appendix A. The aerobic exercise intervention above mentioned was added to TAU.

The TAU intervention group was provided by the Spanish National Health System for FMS patients. This treatment is offered by the general practitioner (GP) and consists of administering pregabalin or other drugs for pain as well as antidepressants such as duloxetine (or another similar SNRI). TAU can also include pharmacological treatments for insomnia and fatigue. Furthermore, GPs may refer the patient to a rheumatologist, psychiatrist, and/or other specialist as required. Psychological treatment is not provided by the Spanish NHS and pharmacotherapy is the frontline option.

### 2.3. Measurements

First, a set of socio-demographic and medication information were assessed (age, age of first illness symptoms, years from first symptoms to diagnosis, length of illness, educational level, work outside home, pain medication, antidepressant medication, and anxiolytic medication). After that, the outcome measures mentioned below were assessed.

#### 2.3.1. Main Outcome

GH responses to physical acute stress (GH 20’): GH is usually stimulated by the stress of vigorous exercise. To evaluate the response of the GH to acute exercise, and after obtaining a basal blood sample, participants carried out the same vigorous physical exercise for 15 min, at sufficient intensity for fatigue to appear. Specifically, since the blood extraction was carried out in a hospital and the sample was taken immediately after the exercise, a secondary staircase of the hospital, close to the extraction room, was used for the participants to go up and down. A slow but constant rhythm of going up and down stairs was used for 15 min. The rhythm was marked by a person from the team, so that 30 steps were going up and down at a regular pace, with the help of the handrail, using between 1.45 and 2 min, and repeating the same action until completing a total of 15 min. The blood sample was taken after waiting for 5 to 10 min.

#### 2.3.2. Secondary Outcomes

Basal blood samples of GH: Serum samples for the hormonal analyses were drawn between 10.00 h. and 10.30 h. The blood samples were obtained from the antecubital vein. Plasma GH was measured by an immunoradiometric assay, with a sensitivity of 0.1 µg/L and with Intrassay coefficients that varied between 6.3% and 4.4% for GH concentrations of 2.2 µg/L and 2.4 µg/L, respectively.

Sensitivity to pain: To assess the variation in the perception of pain at each one of the points described by the ACR criteria, the method applied by Alegre et al. (1995) as cited in [30] was used. This method records the exact point where pain appears, thus permitting verification of variations in the pain threshold between evaluations.

Anthropometric and body composition measurements: The anthropometric variables were taken by an experienced physiotherapist. Weight, sex, body mass index (BMI), abdominal perimeter and adiposity index (sum of the six skinfold measurements of fat) were evaluated. Skinfold fat was measured at the triceps, subscapular, iliac crest, abdominal, front thigh, and medial calf locations using Holtain skinfold callipers [31].

Physical performance: Maximum oxygen uptake (VO_2_ max). Exercise testing to volitional exhaustion (symptom-limited termination) was measured on a cycle ergometer (Ergoline), using a standard protocol of minute stages with 15 watts workload increases beginning with 0 watts for the first minute. The test was suspended when reaching maximum heart rate values, VO_2_ plateau, or any other maximal criterion. The highest value obtained in the last load was taken as peak oxygen uptake (VO_2_ peak). The test was interrupted at the patient’s request because of dyspnea, pain, or muscle fatigue.

Health-related quality of life: The health-related quality of life was assessed by means of The Fibromyalgia Impact Questionnaire (FIQ) and the EuroQol five-dimensional questionnaire (EQ-5D). The FIQ is a self-report inventory that includes 10 items to specifically measure FMS patient’s ability to perform physical activities, the number of days in the last week that they felt good and how many days of work they missed, as well as their ability to work, pain, fatigue, morning tiredness, stiffness, anxiety, and depression. The total score ranges from 0 to 100, and higher scores indicate greater functional difficulties. The Spanish validated version of the FIQ was used, which has demonstrated appropriate psychometric properties [32]. The EQ-5D is a commonly used, easy-to-complete and standardized questionnaire to measure general health-related quality of life. It has become the most widely used instrument because of its brevity and international comparability [33]. The instrument comprises five dimensions of health: mobility, self-care, usual activity, pain/discomfort, and anxiety/depression. The subject describes his or her own health-related quality of life by assigning one of three levels of problem severity (“none”, “some/moderate”, “severe”, coded as 1, 2, and 3, respectively) to each dimension. Combining the five dimensions results in a five-digit number that describes the person’s state of health. Any health state can be converted to a single summary “index score” by using sets of utility values derived from samples of the general population. An index value of 1 represents full health and 0 corresponds with death.

### 2.4. Statistical Analyses

The variables referred above were described using means, standard deviations (SDs), frequencies, and percentages, according to their distribution. The baseline similarity between groups was reviewed using the corresponding t- and χ^2^ tests.

An analysis of covariance (ANCOVA) at post-intervention was developed after assessing assumptions of homocedasticity and normality of outcomes and residuals. The basal scores of the corresponding outcomes were included as co-variates in the analysis, in order to remove their possible influence. The effect size was calculated by means of eta partial squared (Ƞ^2^). Values of <0.01 are considered to be small, of 0.06 as moderate, and of >0.14 as large. The pre-post percentage of change was calculated. The degree of association between the outcomes at pre-intervention and at post-intervention was obtained with the Pearson’s correlation coefficient. The potential predictive role of the pre-intervention levels of GH response to exercise and pre-post improvements in the other outcomes was also explored by means of the Pearson’s correlation coefficient.

The level of significance was set at α < 0.05. Given the exploratory nature of this study, we did not correct for multiple testing [34]. The IBM Statistical Package for the Social Sciences (SPSS) version 29.0 was used to carry out the analysis.

## 3. Results

The study flow chart is reflected in Figure 1.

All the participants suffering from FMS were women. The two groups of participants had baseline similarity in terms of socio-demographic variables (see Table 1). No differences between groups were found at baseline in any of the clinical measurements.

The results of the ANCOVA test (see Table 2) revealed a moderate and significant effect of the treatment based on aerobic exercise in the GH 20’ primary outcome (Ƞ^2^ = 0.07; *p* = 0.048). In addition, a significant and large effect of the treatment based on aerobic exercise was also found in VO_2_ max (Ƞ^2^ = 0.15; *p* = 0.006) and EQ-5D (Ƞ^2^ = 0.10; *p* = 0.019). Moreover, moderate and significant effects were found in the sum of skinfolds (Ƞ^2^ = 0.08; *p* = 0.042). Low effects, although not significant, were found in pressure in tender points (Ƞ^2^ = 0.02; *p* = 0.296), and BMI (Ƞ^2^ = 0.05; *p* = 0.091). Finally, there were no effects of the treatment in GH basal levels (Ƞ^2^ < 0.01; *p* = 0.656) and the FIQ (Ƞ^2^ < 0.01; *p* = 0.807).

With respect to the associations between the variables and the GH 20’ primary outcome (see Table 3), we found that at pre-intervention, GH 20’ was significantly related to GH basal (r = 0.72; *p* < 0.001), VO_2_ max (r = 0.33; *p* = 0.003), BMI (r = −0.41; *p* < 0.001), and sum of skinfolds (r = −0.45; *p* < 0.001). GH basal was significantly related to VO_2_ max, BMI, sum of skinfolds, and EQ-5D. The FIQ was significantly and inversely related to VO_2_ max and sum of skinfolds. The EQ-5D showed this same pattern of relationships but with an inverse sign (the FIQ and the EQ-5D are inversely scored). There were no significant relationships between pressure in tender points and the other variables.

At post-intervention (see Table 4), we observed that the GH 20’ primary outcome was significantly associated with GH basal (r = 0.88; *p* < 0.001), VO_2_ max (r = 0.48; *p* < 0.001), BMI (r = −0.52; *p* < 0.001), sum of skinfolds (r = −0.49; *p* < 0.001), and EQ-5D (r = 0.27; *p* = 0.044). GH basal was also significantly related to VO_2_ max, BMI, and sum of skinfolds. The FIQ measure was significantly and inversely related to VO_2_ max and to pressure in tender points. The EQ-5D measure was significantly related to VO_2_ max.

Interestingly, in the exercise intervention group, the GH 20’ primary outcome at pre-intervention was significantly associated with VO_2_ max (r = 0.37; *p* = 0.047) and FIQ (r = −0.39; *p* = 0.038) pre-post improvements throughout the programme. Finally, we observed that in the exercise intervention group, the more the VO_2_ max increased throughout the programme, the more the response of GH to intense exercise (r = 0.37, *p* = 0.047), and the GH basal (r = 0.38, *p* = 0.040) were enhanced.

## 4. Discussion

It was observed that the intervention carried out in the treated arm, based on high-intensity physical exercise, increased the response of GH to exercise, and the aerobic capacity, body composition and general quality of life were improved. We also observed that initial levels of the GH response to exercise could determine improvements in aerobic capacity and quality of life related to FMS throughout the programme.

We found an unexpected response in the exercise group, as GH basal levels dropped after effort, maybe due to a delay in the response or due to a weak response. This sort of response did not appear in the TAU arm. Despite this, there were no significant between group differences in this regard. There are authors who, in FMS, indicate a late response of this hormone when stimulated. In a study by Riedel et al. [35], patients with FMS suffered a late increase of GH basal, which occurred 60 min after their intravenous stimulation, when its levels had already dropped in healthy individuals. Gürsel et al. [36] find significantly reduced levels of GH basal in women after carrying out intense exercise compared with healthy controls, suggesting a possible alteration in the response of the GH in these patients. There are studies that suggest an abnormality of the response to acute stress in FMS, associating that lack of response to a high basal tone of somatostatin, a GH inhibiting hormone [37]. The high levels of hypothalamic somatostatin could be explained as a response to the corticotropin-releasing hormone, the main hormone involved in the response to stress [38]. On the other hand, nonspecific stress before test could raise basal levels of GH and block the stimulus response. Other possible influencing factor in GH basal response is medication, something that was not controlled in the present study.

According to our results, the exercise programme was helpful in normalizing the response of GH to physical effort. In healthy individuals, physical exercise is a stimulus for the release of the GH hormone [39,40], and in a study by Manetta et al. [41], the GH response to the exercise in trained individuals was three times greater than in sedentary individuals. Physical exercise plays an important role in the regulation of the HPA axis [39]. Appropriate physical exercises could be used to modify the levels of some stress-related hormones. The baseline relationship between VO_2_ max and basal GH levels in women with FMS suggest that those women who have a better physical condition, have more normalised hormone parameters. Moreover, the relationship found between the improvement of VO_2_ max and the increase of the GH response to intense physical activity in the exercise group, as well as the potential moderating role that the initial levels of GH response to intense physical activity might play to take advantage of exercise programmes, supports the importance of physical conditioning in the normalization of neuro-hormonal and quality of life parameters in patients suffering from FMS.

The reduction of the sum of skinfolds in the FMT after the intervention indicates an important change in body composition. During the first evaluation, we found a negative relationship in participants with FMS between basal levels of GH and umbilical perimeter, and likewise between sum of skinfolds and basal levels of GH. An elevated risk of both abdominal and visceral adiposity has been established in women with GH deficiency caused by hypopituitarism, and decreased GH secretion has been implicated as a risk factor for abdominal and visceral obesity [42]. The review work by Jones et al. [11] on alterations of the GH in FMS already indicated the need to incorporate associations between abdominal adiposity and GH into the analyses, as this is usually associated with a reduction in the secretion of this hormone [43]. These findings point to the possible relationship that exists in FMS between body composition and GH levels. Thus, we believe that an improvement in body composition based on an appropriate physical exercise programme may contribute to normalising the behaviour of the GH. Nevertheless, we also found decreased BMI levels in the TAU group because of an unexplained weight loss. Finally, the improvement of VO_2_ max and general quality of life in the exercise group ratify the results of previous works [44,45], and support the crucial role that physical exercise may play in the treatment of FMS, and the relationships between physical and mental health.

### Strengths and Limitations

There are not many studies that analyse the effect of physical exercise programmes on FMS patients and that take hormonal variables as main outcome measures, and the existing ones do not deepen into key aspects about the influence of exercise on the GH, either. However, we recognise that only having one measurement to determine the basal levels of GH is a limitation, as it has shown a pulsatile secretion [11], which would make collection over 24 h more reliable. As a strength we stress that, despite this being a high intensity exercise programme, it had a low rate of abandonment in the treatment group, as there was only one which was caused by the exacerbation of symptomatology over the first weeks. Thus, and due to the improvements obtained in the outcome measures, it seems that a high intensity physical exercise programme can be tolerated and has great benefits in women with FMS, providing that it is supervised and controlled. These benefits could be more associated with quality-of-life aspects related to FMS than with changes in sensitivity to pain. Nevertheless, this is an exploratory and hypotheses generating study that needs further replications using pre-registered research to better understand the role of the GH and the way through which this type of programmes could produce improvements in the quality of life of FMS patients.

## 5. Conclusions

Monitoring a low impact aerobic-based, high intensity, physical exercise programme may increase the response of the GH to exercise, improve VO_2_ max, body composition, and general quality of life in women with FMS. It seems this type of programmes could be well-tolerated if they are supervised and controlled. It is important to normalise the response of the GH to exercise, which could be related to a series of beneficial aspects [46]. It is relevant to highlight the relationships found in FMS patients between GH levels, aerobic capacity and body composition, and the importance of physical conditioning in the normalization of neuro-hormonal parameters and other related variables. Although it is difficult to establish causal relationships, the normalization of neuro-hormonal patterns could be a keystone factor for improvements in the FMS symptomatology. The behaviour of the GH, as well as its potential relationships among anthropometric measures, aerobic capacity, and quality of life, must be studied in greater depth in future works.

## Figures and Tables

**Figure 1 ijerph-20-02261-f001:**
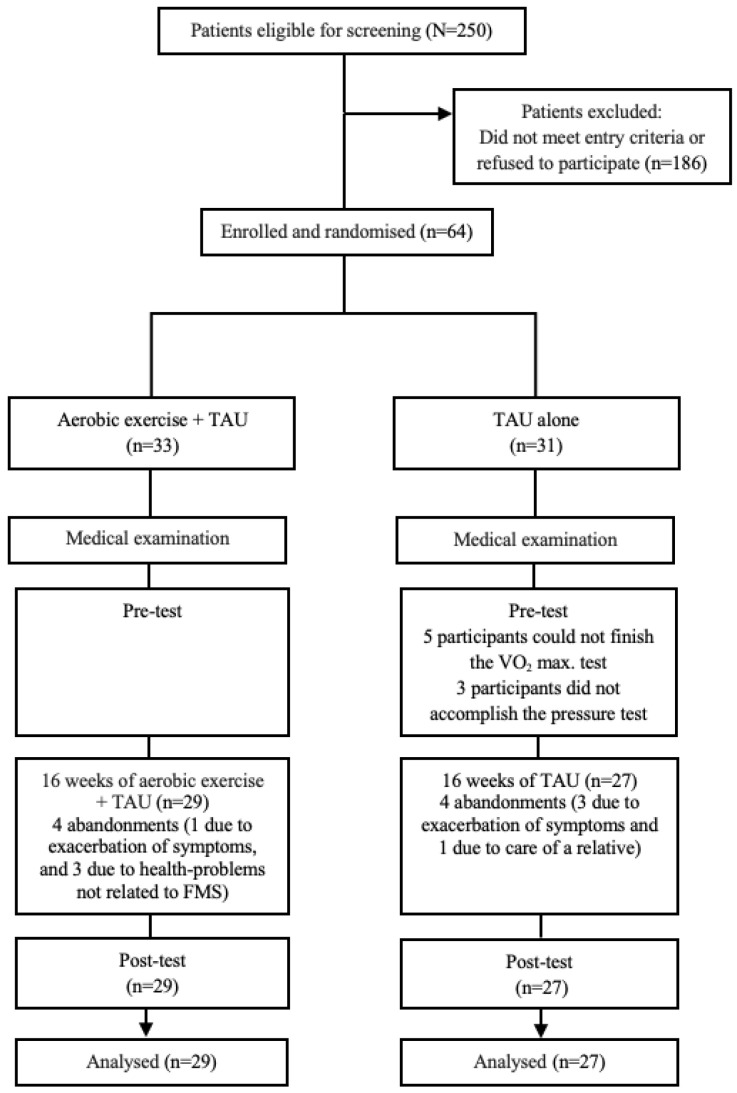
Flowchart of participants in the randomized controlled trial.

**Table 1 ijerph-20-02261-t001:** Baseline sociodemographic characteristics of patients by group.

	Exercise (n = 29)	TAU (n = 27)	*p*
Sociodemographic and Medication			
Age, years, M (SD)	46.9 (9.7)	48.7 (7.8)	0.465
Age of first illness symptoms, years, M (SD)	30.1 (12.8)	32.7 (11.9)	0.371
Years from first symptoms to diagnosis, M (SD)	13.3 (12.5)	11.8 (10.4)	0.805
Length of illness, years, M (SD)	16.8 (13)	15.8 (10.8)	0.954
Educational level, frequency (%)			0.950
No studies/primary	14 (48.3)	13 (48.1)	
Secondary studies	8 (27.6)	8 (29.6)	
University	7 (24.1)	6 (22.2)	
Work outside home, yes, frequency (%)	23 (79.3)	21 (77.8)	0.890
Pain medication, yes, frequency (%)	16 (55.2)	13 (48.1)	0.599
Antidepressant medication, yes, frequency (%)	14 (48.3)	11 (40.7)	0.571
Anxiolytic medication, yes, frequency (%)	13 (44.8)	10 (37.0)	0.554

Note. M = mean. SD = standard deviation.

**Table 2 ijerph-20-02261-t002:** Analysis (ANCOVA) of the primary and secondary outcomes.

	Exercise (n = 29)			TAU (n = 27)				
Variables	Pre-	Post-	Δ%	Pre-	Post-	Δ%	*p*	Ƞ^2^
GH 20’ (ng/mL)	2.87 (2.88)	4.74 (6.00)	65.2	3.76 (4.84)	3.38 (5.21)	−10.1	0.048	0.07
GH (ng/mL)	3.65 (3.89)	3.45 (4.57)	−5.5	2.40 (2.72)	2.60 (4.99)	8.3	0.656	<0.01
VO_2_ Max. (ml/kg/min) ^a^	20.06 (5.58)	22.02 (5.68)	9.8	19.18 (4.60)	18.47 (4.17)	−3.7	0.006	0.15
Pressure (cm^3^) ^b^	5.68 (2.58)	6.53 (2.68)	15.0	5.58 (2.70)	5.94 (2.17)	6.5	0.296	0.02
BMI (kg/m^2^)	26.71 (3.90)	26.64 (4.15)	−0.3	27.03 (5.89)	26.43 (5.66)	−2.2	0.091	0.05
Σ skinfolds	156.78 (43.77)	141.23 (39.39)	−9.9	153.41 (51.65)	149.37 (42.72)	−2.6	0.042	0.08
FIQ	66.90 (14.44)	58.71 (19.08)	−12.4	75.81 (14.08)	68.36 (20.44)	−9.8	0.807	<0.01
EQ-5D	0.22 (0.29)	0.40 (0.34)	81.8	0.21 (0.31)	0.23 (0.31)	8.7	0.019	0.10

Note. Scores in pre- and post- are means (SDs). Δ% = increase percentage. ^a^ Five participants could not finish the Max.VO_2_ test in the TAU group. ^b^ Three participants did not accomplish the pressure test in the TAU group.

**Table 3 ijerph-20-02261-t003:** Pre-intervention correlations between outcomes using the total sample.

	1	2	3	4	5	6	7
1. GH 20’							
2. GH basal	0.72 ***						
3. VO_2_ max	0.33 **	0.47 ***					
4. Pressure	0.19	0.19	0.16				
5. BMI	−0.41 ***	−0.50 ***	−0.50 ***	0.01			
6. Σ skinfolds	−0.45 ***	−0.49 ***	−0.38 **	0.15	0.83 ***		
7. FIQ	−0.11	−0.19	−0.34 **	−0.18	0.07	0.27 *	
8. EQ-5D	0.15	0.23 *	0.34 **	0.27	−0.14	−0.30 *	−0.87 ***

Note. *** *p* < 0.001; ** *p* < 0.01; * *p* < 0.05.

**Table 4 ijerph-20-02261-t004:** Post-intervention correlations between outcomes using the total sample.

	1	2	3	4	5	6	7
1. GH 20’							
2. GH basal	0.88 ***						
3. VO2 max	0.48 **	0.38 **					
4. Pressure	0.17	0.20	0.22				
5. BMI	−0.52 ***	−0.50 ***	−0.41 **	−0.12			
6. Σ skinfolds	−0.49 ***	−0.47 ***	−0.40 **	0.05	0.79 ***		
7. FIQ	−0.20	−0.15	−0.40 **	−0.38 **	0.10	0.18	
8. EQ-5D	0.27 *	0.22	0.43 **	0.19	−0.07	−0.21	−0.79 ***

Note. *** *p* < 0.001; ** *p* < 0.01; * *p* < 0.05.

## Data Availability

The data that support the findings of this study are available from the corresponding author upon reasonable request.

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
