# Peer review of "Can an Aerobic Exercise Programme Improve the Response of the Growth Hormone in Fibromyalgia Patients? A Randomised Controlled Trial"

_ijerph, 2023, doi:10.3390/ijerph20032261_

Round 1

Reviewer 1 Report

please see attached!

Author Response

The only request I would have from the researchers is to present the strength training exercises that the fibromyalgia patients followed. That may be provided as a supplementary file or just add the exercises that they performed.

Answer: Many thanks for this suggestion. We have now included in the manuscript the following supplement:

Supplement 1: Details of the training programme

The training program included integrated work on aerobic endurance and dynamic strength. The sessions were mainly based on low-impact aerobic dance (lower intensity, lower risk of injury, and less repercussions at the joint level than high-impact dance), combined with strength exercises with self-loads, and relaxation techniques for the 'back-to-the-calm' phase. The programme was adapted to the ACSM guidelines (Busch et al., 2007), with the sessions following the same structure:

  1. Warm-up (10-15 minutes): Locomotion exercises, including joint mobility work, always respecting comfortable ranges of motion for each participant.

  1. Main part (30-35 minutes):

2.1. Cardiovascular endurance work through low-impact aerobic dance, including mainly global and locomotion movements, involving both the lower and upper body.

2.2 Strength endurance work, through calisthenics, performing a total of 6 to 7 exercises, with 15 to 25 repetitions each, as established by the ACSM guidelines to improve muscular endurance (Pescatello, 2014). Exercises included squat, forward lunges, shoulder bridge, chest lift, superman, bicycle in air, top leg abduction, etc.

  1. Back-to-the-calm (10-15 minutes): Relaxation techniques and breathing exercises.

The intensity of the sessions was controlled through pulsometers, registering an average heart rate in the aerobic part of around 140 BPM, which would correspond to values close to 80% of the Maximum Heart Rate, considering the mean age of programme participants for its establishment. As soon as the session finished, the Borg scale (Borg, 1976) was passed to verify if the training progression was adequate.

References

Borg G. Simple rating methods for estimation of perceived exertion. Wenner-Gren Center International Symposium 1976. Series 28:39-47.

Busch AJ, Barber KA, Overend TJ, et al. Exercise for treating fibromyalgia syndrome. Cochrane Database Syst Rev. 2007; CD003786. doi: 10.1002/14651858.CD003786.pub2.

Pescatello L. In: ACSM’s Guidelines for Exercise Testing and Prescription. 9th ed. Pescatello L.S., Arena R., Riebe D., Thompson P.D., editors. Volume 58. ASCM Group Publisher; Baltimor, MD, USA: Philadelphia, PA, USA: 2014. p. 328. The Journal of the Canadian Chiropractic Association.

Reviewer 2 Report

I would like to congratulate the authors on this study, I think it is novel and the topic is really interesting. 

My comments are as follows;

1) is there a study protocol for this RCT that was previously published. If not, could you explain why you did not seek to publish a study protocol prior to running this trial?

2) Why did you expect a maximum attrition rate of 20%? was this just an estimated guess or is there a reason why this high amount of attrition would be expected? Perhaps you could include a sentence to explain this in your paper since I think this work may motivate some researchers to do similar studies. 

3) Under the interventions section. I think you need to include a new table outlining the exact exercises ( strength, aerobic etc) in the intervention group. Either include this as a table or supplementary attached file. I think you need to show an example of the routines used in addition to stating you followed ACSM guidelines. Since it was group based intervention I would assume that each participant did not have an individualized program, but used loads relative to their physical ability. Please include as it will be a nice addition for the reader. 

4) Can you clarify what specific exercise you used in the GH test for 15 minutes?

5) I think it would be best to include your LIMITATIONS section, as a separate section vs just including them in a discussion.

Overall this is a good study and should be very interesting to readers. I look forward to the final version 

Author Response

1) is there a study protocol for this RCT that was previously published. If not, could you explain why you did not seek to publish a study protocol prior to running this trial?

Answer: Thanks for this comment. There is not a study protocol due to the very exploratory nature of the present study, as it is stated in pages 5 and 9. The results of the present study will nevertheless facilitate the adequate design of future studies that will test specific hypotheses with their corresponding registration. We have now recognized (page 9) that: "Nevertheless, this is an exploratory and hypotheses generating study that needs further replications using pre-registered research to better understand the role of the GH and the way through which this type of programmes could produce improvements in the quality of life of FMS patients".

2) Why did you expect a maximum attrition rate of 20%? was this just an estimated guess or is there a reason why this high amount of attrition would be expected? Perhaps you could include a sentence to explain this in your paper since I think this work may motivate some researchers to do similar studies.

Answer: Thanks for this suggestion. We have now included in the manuscript that: "In line with previous research in which around 85% of participating women with fibromyalgia tolerated a moderate to high–intensity aerobic exercise programme [X], we expected a maximum attrition rate in our low impact aerobic-based, high intensity, physical exercise programme of around 20%, and thus numbers were increased to obtain a total sample size of 64 participants".

References

Mannerkorpi K, Nordeman L, Cider A, Jonsson G. Does moderate-to-high intensity Nordic walking improve functional capacity and pain in fibromyalgia? A prospective randomized controlled trial. Arthritis Res Ther. 2010;12:R189. doi: 10.1186/ar3159.

3) Under the interventions section. I think you need to include a new table outlining the exact exercises (strength, aerobic etc) in the intervention group. Either include this as a table or supplementary attached file. I think you need to show an example of the routines used in addition to stating you followed ACSM guidelines. Since it was group-based intervention, I would assume that each participant did not have an individualized program, but used loads relative to their physical ability. Please include as it will be a nice addition for the reader.

Answer: Many thanks for this suggestion. We have now included in the manuscript the following supplement:

Supplement 1: Details of the training programme

The training program included integrated work on aerobic endurance and dynamic strength. The sessions were mainly based on low-impact aerobic dance (lower intensity, lower risk of injury, and less repercussions at the joint level than high-impact dance), combined with strength exercises with self-loads, and relaxation techniques for the 'back-to-the-calm' phase. The programme was adapted to the ACSM guidelines (Busch et al., 2007), with the sessions following the same structure:

  1. Warm-up (10-15 minutes): Locomotion exercises, including joint mobility work, always respecting comfortable ranges of motion for each participant.
  2. Main part (30-35 minutes):

2.1. Cardiovascular endurance work through low-impact aerobic dance, including mainly global and locomotion movements, involving both the lower and upper body.

2.2 Strength endurance work, through calisthenics, performing a total of 6 to 7 exercises, with 15 to 25 repetitions each, as established by the ACSM guidelines to improve muscular endurance (Pescatello, 2014). Exercises included squat, forward lunges, shoulder bridge, chest lift, superman, bicycle in air, top leg abduction, etc.

  1. Back-to-the-calm (10-15 minutes): Relaxation techniques and breathing exercises.

The intensity of the sessions was controlled through pulsometers, registering an average heart rate in the aerobic part of around 140 BPM, which would correspond to values close to 80% of the Maximum Heart Rate, considering the mean age of programme participants for its establishment. As soon as the session finished, the Borg scale (Borg, 1976) was passed to verify if the training progression was adequate.

References

Borg G. Simple rating methods for estimation of perceived exertion. Wenner-Gren Center International Symposium 1976. Series 28:39-47.

Busch AJ, Barber KA, Overend TJ, et al. Exercise for treating fibromyalgia syndrome. Cochrane Database Syst Rev. 2007; CD003786. doi: 10.1002/14651858.CD003786.pub2.

Pescatello L. In: ACSM’s Guidelines for Exercise Testing and Prescription. 9th ed. Pescatello L.S., Arena R., Riebe D., Thompson P.D., editors. Volume 58. ASCM Group Publisher; Baltimor, MD, USA: Philadelphia, PA, USA: 2014. p. 328. The Journal of the Canadian Chiropractic Association.

4) Can you clarify what specific exercise you used in the GH test for 15 minutes?

Answer: Thanks for this suggestion. We have now included the following:

Specifically, since the blood extraction was carried out in a hospital, and the sample was taken immediately after the exercise, a secondary staircase of the hospital, close to the extraction room, was used for the participants to go up and down. A slow but constant rhythm of going up and down stairs was used for 15 minutes. The rhythm was marked by a person from the team, so that 30 steps were going up and down at a regular pace, with the help of the handrail, using between 1.45 and 2 minutes, and repeating the same action until completing a total of 15 minutes.

5) I think it would be best to include your LIMITATIONS section, as a separate section vs just including them in a discussion.

Asnwer: Thanks for this suggestion. We have now included the following sub-section in the discussion: "4.1. Strengths and limitations".